# The Influence of Catechols on the Magnetization of Iron Oxide Nanoparticles

**DOI:** 10.3390/nano13121822

**Published:** 2023-06-08

**Authors:** Stanislav Čampelj, Matic Pobrežnik, Tomas Landovsky, Janez Kovač, Layla Martin-Samos, Vera Hamplova, Darja Lisjak

**Affiliations:** 1“Jožef Stefan” Institute, Jamova 39, 1000 Ljubljana, Slovenia; janez.kovac@ijs.si (J.K.); darja.lisjak@ijs.si (D.L.); 2CNR-IOM, Democritos National Simulation Center, Istituto Officina dei Materiali, c/o SISSA, Via Bonomea 265, 34136 Trieste, Italy; matic.pobreznik@ijs.si (M.P.);; 3FZU—Institute of Physics of the Czech Academy of Science, Na Slovance 1999/2, 18200 Prague, Czech Republic; landovsky@fzu.cz (T.L.); hamplova@fzu.cz (V.H.)

**Keywords:** magnetic nanoparticles, catechols, adsorption, magnetic properties, XPS, DFT

## Abstract

In this study, MNPs were functionalized with pyrocatechol (CAT), pyrogallol (GAL), caffeic acid (CAF), and nitrodopamine (NDA) at pH 8 and pH 11. The functionalization of the MNPs was successful, except in the case of NDA at pH 11. The thermogravimetric analyses indicated that the surface concentration of the catechols was between 1.5 and 3.6 molecules/nm^2^. The saturation magnetizations (M_s_) of the functionalized MNPs were higher than the starting material. XPS analyses showed only the presence of Fe(III) ions on the surface, thus refuting the idea of the Fe being reduced and magnetite being formed on the surfaces of the MNPs. Density functional theory (DFT) calculations were performed for two modes of adsorption of CAT onto two model surfaces: plain and adsorption via condensation. The total magnetization of both adsorption modes remained the same, indicating that the adsorption of the catechols does not affect the M_s_. The analyses of the size and the size distribution showed an increase in the average size of the MNPs during the functionalization process. This increase in the average size of the MNPs and the reduction in the fraction of the smallest (i.e., <10 nm) MNPs explained the increase in the M_s_ values.

## 1. Introduction

For more than 20 years, superparamagnetic iron oxide nanoparticles (MNPs) have been attracting a lot of attention for different applications, from medicine to sensors [1,2,3,4,5,6,7,8]. For all applications, their surface properties must be tailored to meet specific criteria, such as stability in different solvents and, in the case of medical use, biocompatibility. For this purpose, the MNPs are functionalized with different ligands. The contribution of these ligands to the saturation magnetization (M_s_) is negligible when compared to the M_s_ of magnetic materials, such as with MNPs. For clarity, we refer to the studied MNPs as magnetic materials and to the paramagnetic/diamagnetic ligands as non-magnetic materials. When the MNPs are coated with inert ligands, the M_s_ is reduced by the addition of a non-magnetic material. Magnetization is expressed as the magnetic moment per unit of mass. Any addition of non-magnetic material will only increase the mass of the sample without any increase in the magnetic moment. In contrast, “non-innocent” ligands, such as catechols or phosphonates, affect the M_s_ of the MNPs, as was reported in several studies [1,2,9]. There are two explanations for how the catechols increase the M_s_ of MNPs: either due to a reduced surface-spin canting [1,2,10,11] or a reduction in the surface Fe(III) to Fe(II) [2,7,12,13].

The crystal structure on the surface of the MNPs is disrupted relative to the crystal structure in the core. The surface atoms are under-co-ordinated. Therefore, the magnetic spins in the surface layer are not aligned (they are canted) [1,2,10,11] with the magnetic spins in the core of the material, forming a magnetically dead layer. The reported thickness for this layer is typically ~1 nm. With decreasing particle size, the fraction of surface atoms in the dead layer increases. As a result, the M_s_ of small nanoparticles is less than for larger ones. According to [2,8,10,14,15,16,17], catechols adsorbed on the surface influence the surface structure of MNPs. The reason for the change in the structure of the outer layer is the changed Fe–O bond length. The Fe–O bond length in [Fe(cat)3]^3−^ is similar to the Fe–O bond lengths at the octahedral sites in magnetite, i.e., 2.017 Å and 2.058 Å, respectively [2]. The structure of the outer layer becomes more similar to the bulk structure when coordinated with catechols. The consequence is an alignment of the spins in the surface layer, and so the magnetization of the MNPs increases [1].

Catechols are known for their ability to reduce Fe(III) to Fe(II) [12,13], while catechol is oxidized to semiquinone or quinone. Similarly, there is a possibility to transform the surface layer from maghemite to magnetite. Since the M_s_ of the magnetite is higher than the M_s_ of maghemite, the overall M_s_ of the MNPs increases.

Our aim was to determine the influence of pH on the adsorption of different catechols on the MNPs and to elucidate the origin of the increased M_s_ of the functionalized MNPs. The amount of adsorbed catechols on the surfaces of the metal oxide particles increases at higher pH values [17]. Depending on which pH the catechols are adsorbed onto the surfaces of metal oxides, either mono- or bidentate bonds are formed [5,18,19,20]. The formation of a complex between the ligands and the iron, and therefore the adsorption of catechols on the MNPs, is related to the pK_a_ of the hydroxyl groups [21]. The pK_a_ of the hydroxyl groups changes when an electron-withdrawing or electron-donating group is introduced to the benzene ring. For comparison, we used catechols with different functionalization groups. The chosen catechols were as follows: (Figure 1): pyrocatechol (CAT), which was used as a reference point since it has only two OH groups without any additional electron donor or acceptor groups; pyrogallol (GAL), which has an additional OH group that increases the electron density in the benzene ring in comparison to CAT; caffeic acid (CAF), which has an electron-acceptor carboxyl group that decreases the electron density; and nitrodopamine (NDA), which has an electron-acceptor nitric group that also decreases the electron density. NDA also has an amine group at the end of an alkyl chain. Since electrons can only move via π bonds or lone pairs, the amine group does not affect the electron density in the catechol. NDA was used in the form of sulfuric salt to increase its solubility in water. The pK_a_ values for the dissociation of the first OH group are typically around 9, and for the dissociation of the second OH group, above 11: Table 1 [4,22,23].

## 2. Materials and Methods

### 2.1. Experimental

The MNPs were synthesized by coprecipitating Fe(II) and Fe(III) ions with a concentrated ammonia solution in a two-step process, as described in [24]. In the first step, the pH value of the solution was increased and maintained at pH = 3 for 30 min with a diluted ammonia solution. During the first step, iron hydroxides precipitate. In the second step, the pH was sharply raised to pH = 11.6 with a concentrated ammonia solution. In this step, iron hydroxide was oxidized by oxygen, thus forming a spinel product. The suspension of MNPs was aged for 30 min to allow all the total oxidization Fe(II) to Fe(III). The MNPs were washed with a diluted ammonia solution with a pH above 10. After the synthesis, the MNPs were functionalized with catechols. In a typical reaction, the concentration of MNPs in water was set at 5 mg/mL. The pH was set to pH = 8 or pH = 11, and the catechols were added. The concentration of the catechols was set to 10 molecules/nm^2^ of particles. The flask with the as-prepared suspension was placed in an ultrasonic bath for 2 h. After that, the particles were centrifuged several times in a high-performance centrifuge (Sorvall Lynx 6000 Superspeed Centrifuge, Thermo Fisher Scientific Inc., Waltham, MA, USA) at 50,000× *g* for 15 min. After each centrifugation, the supernatant was discarded, and the particles were redispersed in deionized (DI) water. For each pH value, a control test was carried out in the same manner but without the addition of catechols.

### 2.2. Characterization

Several techniques were used to characterize the samples.

#### 2.2.1. Electro-Kinetic Measurements

Electrokinetic measurements were performed using (Litesizer 500, Anton Paar, Graz, Austria) particle size analyzer. The zeta-potentials of bare and functionalized MNPs in aqueous dispersions were measured as a function of pH. The pH of the suspensions was adjusted using solutions of NaOH and HCl in the range from pH 2 to pH 11. The shift of the isoelectric point is an indication of the successful functionalization of the MNPs.

#### 2.2.2. Thermogravimetry

The decomposition temperature of the pure catechols in air was confirmed with thermogravimetric analysis (TGA) using (TG-DTA Setsys Evolution, Setaram, Caluire-et-Cuire, France). The TGA results showed that the catechols decomposed completely below 600 °C. Based on the results from the TGA, the samples were fired in a furnace at 650 °C in air, with a heating rate of 2.5 K/min to determine the number of adsorbed species. The samples were kept at 650 °C for 60 min to ensure the complete decomposition of the catechols.

#### 2.2.3. Magnetic Measurements

The room-temperature magnetic properties of dried samples were measured with a vibrating-sample magnetometer (VSM 7407, Lake Shore Cryotronics Inc., Westerville, OH, USA). The magnetic properties were measured in a continuous loop in the range from −10,000 Oe to 10,000 Oe with an increment of 200 Oe.

#### 2.2.4. Transmission Electron Microscopy

The sizes and size distributions of the bare and functionalized MNPs were determined using a transmission electron microscope (TEM, Jeol 2100, with EDXS spectrometer JED 2300 EDS, Tokyo, Japan). The images taken with the TEM were processed using Digital Micrograph^TM^, Gatan Inc., Plaeston, CA, USA. Individual MNPs were marked manually. The minimum number of marked particles was 250. The sizes and size distributions of the particles were determined using the log-normal function.

#### 2.2.5. XPS

The samples were analyzed using a TFA XPS spectrometer (Physical Electronics Inc., Chanhassen, MN, USA) equipped with a monochromated Al-Kα X-ray source under ultra-high vacuum. The high-energy-resolution spectra of the characteristic peaks for elements Fe 2p, Fe 3p, O 1s, N 1s, and C 1s were recorded over a narrow energy range. The spectra were analyzed using MultiPak, Physical Electronics Inc., Chanhassen, MN, USA.

#### 2.2.6. ICP-OES

The concentration of dissolved iron after the adsorption of NDA at pH 8 was determined with an inductively coupled plasma spectrometer (Agilent 5800 ICP-OES VDV, Santa Clara, CA, USA).

### 2.3. Computational Approach

The surfaces of the MNPs were modelled within the framework of the density functional theory (DFT), employing the Hubbard U correction (DFT + U) [25], as implemented in the Quantum ESPRESSO package [26,27,28] for electronic structure calculations. Specifically, the functional PBE [29] was employed in combination with a plane-wave basis set and ultra-soft pseudo-potentials [30,31]. The kinetic energy cut-off for the wave functions was set to 50 Ry, whereas a 500 Ry cut-off was applied in terms of the charge density. Additionally, the dispersion correction of Grimme D3 [32] was employed to account for the lateral interactions among the molecules forming the adsorbed layer on the surface. To construct the surfaces of the MNPs, two crystalline phases of iron oxides were considered, i.e., hematite (α-Fe_2_O_3_) and magnetite (Fe_3_O_4_). The reasons for this choice are discussed in greater detail in the Surface Models subsection.

Hematite crystallizes in a hexagonal structure with experimental lattice parameters of a = 5.01 Å and c = 13.65 Å [33]. We calculated the Hubbard U parameter from first principles [34]. The parameter was calculated for the rhombohedral unit cell of hematite by employing a 5 × 5 × 5 uniform Monkhorst-Pack k-mesh [35]. We obtained U values of 3.54 eV for the majority spin Fe ions and 3.50 eV for the minority spin Fe ions. The optimum lattice parameters were then calculated by performing volume cell minimizations, from which we obtained a = 5.15 Å and c = 13.93 Å, which is in reasonable agreement with the experimental values. Hematite is antiferromagnetic; hence we obtained a total magnetization of 0 μbohr/cell. On the other hand, magnetite crystallizes in a cubic structure, with a lattice parameter of 8.38 Å. The U values were obtained similarly to those for hematite, i.e., they were calculated in a rhombohedral unit cell employing a 3 × 3 × 3 uniform k-mesh, and we obtained U = 3.83 eV for the majority spin Fe ions and U = 3.56 eV for the minority spin Fe ions. By setting these values for U, we obtained a lattice parameter of 8.55 Å. Maghemite is ferrimagnetic, and we obtained the expected magnetization of 32 μbohr/cell.

#### 2.3.1. Surface Models

The exact surface termination of the synthesized MNPs is not known. In fact, the preferential surface termination of crystalline maghemite is still under debate (see [36] and references therein). Among the plausible surface reconstructions, it has been proposed that the surface layer of γ-Fe_2_O_3_(111) displays either a magnetite-like surface structure under reduced conditions or an α-Fe_2_O_3_-like surface layer under oxidizing conditions [37]. Thus, in order to keep in line with these findings, the adsorption of CAT was studied on the Fe_3_O_4_(111) surface and on the α-Fe_2_O_3_(0001) surface. The bulk structures were cut along the (111) direction for Fe_3_O_4_ and the (0001) direction for α-Fe_2_O_3_(0001). Given that MNPs are exposed to the atmosphere (or they are in solution), the surfaces are assumed to be hydroxylated, and the fully hydroxylated oxygen-terminated surfaces were chosen as being representative. Their structures (top and side views) are shown in Figure 2. The periodic (symmetric) slabs were separated by at least 15 Å of vacuum in the z direction. We employed a 5 × 5 × 1 uniform k-mesh for the unit cell of α-Fe_2_O_3_(0001) and a 4 × 4 × 1 for the unit cell of Fe_3_O_4_(111).

#### 2.3.2. Adsorption Modes

To elucidate the effect (or lack thereof) of the catechols on the magnetic moment of the MNPs, we chose to model two adsorption modes of pyrocatechol, i.e., plain adsorption and adsorption via condensation. Plain adsorption is achieved mainly through hydrogen bonds and does not involve a chemical reaction:CATH + * → CATH*,
where * denotes an empty adsorption site, CATH denotes the intact pyrocatechol, whereas CATH* denotes the pyrocatechol adsorbed onto the surface. On the other hand, adsorption via condensation involves a chemical reaction with a surface OH group, resulting in the formation of a water molecule and the creation of a strong chemical bond to the surface. It can be written as
CATH + OH* → CAT* + H_2_O, 
where OH* represents a surface hydroxyl group, whereas CAT* represents the pyrocatechol adsorbed onto the surface (hence, with one less hydrogen atom), and the formed water molecule is assumed to be released into the gas phase. The reaction energies are therefore calculated as
∆*E_ads_* = *E_CATH_*_*_ − *E_slab_* − *E_CATH_*,(1)
and
∆*E_ads_* = *E_CAT_*_*_ + *E_H2O_* − *E_slab_* − *E_CATH_*,(2)
where *E_CATH_*_*_ and *E_CAT_*_*_ represent the total energies of the respective adsorbed species, *E_slab_* is the total energy of the pristine surface, whereas *E_CATH_* and *E_H2O_* represent the total energies of the pyrocatechol and water molecule in the gas phase.

## 3. Results and Discussion

Figure 3a shows a TEM image of bare MNPs with selected-area electron diffraction (SAED). From the SAED, it is clear that the MNPs have a cubic structure corresponding to maghemite. The hysteresis loop, Figure 3b, reveals that the MNPs are superparamagnetic with M_s_ = 57.1 emu/g.

The samples prepared as described above were characterized using electrokinetic measurements, i.e., zeta potential. The surfaces of the bare MNPs in water are hydrated, resulting in a layer of hydroxyl groups [18]. At pH values of around 7, the MNPs have a neutral charge, i.e., the net charge of the adsorbed species is zero. From the shift of the isoelectric point (IEP), we can see the change in the surface charge of the MNPs and deduce whether the functionalization was successful. The IEP of the functionalized MNPs shifts to an acidic or basic pH, depending on the functionalization group of the used catechol. The carboxylic group in CAF is expected to shift the IEP towards acidic pH values since the deprotonation of the carboxylic occurs at acidic pH values (Table 1). The amine group in the NDA undergoes deprotonation at a pH above 9 and is expected to shift the IEP to higher pH values.

The zeta potential measurements of the MNPs functionalized at pH = 8, Figure 4a, show the expected shifts in IEP for the used catechols. The IEP for the particles functionalized with NDA shifted to pH = 9.2, indicating that the catechols bonded with OH groups, while the alkyl chain with the amine group was oriented outwards. Similarly, the MNPs were successfully functionalized with CAF. The IEP shifted to pH = 4, indicating that the carboxyl group was oriented outwards. The MNPs functionalized with CAT and GAL had IEPs at pH = 5.3 and pH = 3.2, respectively. The OH groups introduced with catechols increased the negative charge on the MNP surfaces [38]. The more OH groups we introduced, the higher the negative charge on the surface. Similar shifts in IEPs can be observed when the catechols were adsorbed at pH = 11, Figure 4b. When the MNPs were functionalized with GAL and CAF at pH = 11, a similar shift in IEP towards the acidic regions, pH = 3.5 and pH = 4.2, respectively, took place. When functionalization with CAT took place at pH 11, the IEP shifted to pH = 4.5. This suggests that the density of the OH groups on the surfaces of the MNPs increased when the reaction took place at pH = 11, compared with pH = 8. However, in the case of functionalization with NDA, the IEP is almost the same as with the unfunctionalized MNPs. This is an indication that the NDA either did not adsorb onto the surface of the MNPs or that it adsorbed in a multilayer.

The mass fraction of the catechols in the samples was determined thermogravimetrically, Table 2. The mass loss for the pure MNPs was about 5% due to adsorbed water and other adsorbed gases. The mass loss for the rest of the samples was between 4 and 6.4%, except for the MNP-NDA-pH11, with a huge mass loss of 22.7%. Based on the mass-loss results, we estimated the coverage of the surfaces of the MNPs with catechols. The number of molecules adsorbed onto the surfaces of the MNPs varies from 1.5 to 3.6 molecules/nm^2^. The values are in good agreement with the calculated values for a monolayer of adsorbed catechols when the catechols form chemical bonds with the surfaces of the MNPs. In contrast, the high mass fraction of MD-NDA-pH11 of 22.7% (or almost 10 molecules/nm^2^) is a strong indication that the NDA adsorbed in a multilayer instead of a monolayer. This would also explain the IEP at neutral pH. The outer layer consists of an equal amount of positively and negatively charged species, leading to a neutral net surface charge.

The measured saturation magnetizations (“M_s_ meas”) are presented in Table 2. The “M_s_ meas” for the functionalized particles were higher than for the bare MNPs, with the exception of the MNPs functionalized with NDA at pH 11 (MNP-NDA-pH11). Since M_s_ is the magnetic moment divided by the mass of the sample, we would expect a decrease in magnetization after the functionalization of the MNPs with catechols. The change in M_s_ is more apparent when we estimate the M_s_ of pure MNPs (“M_s_ pure”). The “M_s_ pure” were recalculated based on the mass loss determined by thermogravimetry from “M_s_ meas”. The “M_s_ pure” of the starting MNPs is 60.0 emu/g, which is in accordance with the reported M_s_ for maghemite [39]. The M_s_ values for all the other samples were 3–13% higher than for the starting MNPs.

One possible explanation for the increase in M_s_ is the reduction of Fe(III) to Fe(II) and the formation of magnetite [12,13]. Fe(II) ions should exhibit an XPS signal at 53.7 eV, while Fe(III) ions should exhibit an XPS signal at 55.6 eV. The largest increase in magnetization was observed for the MNPs functionalized with NDA at pH = 8. Therefore, if a REDOX reaction took place, the Fe(II) concentration would be expected to be the highest in this case. For this purpose, the starting MNPs and the MNPs functionalized with NDA at pH 8 (i.e., the MNPs with the largest increase in M_s_) were investigated with XPS. The Fe(III)/Fe(II) ratio was estimated from the Fe 3p spectrum, Figure 5, which is the most suitable for the determination of the Fe(III)/Fe(II) ratio [40,41].

The XPS analysis of the MNPs only shows a peak at 55.6 eV. The lack of Fe(II) signal confirms the starting material was maghemite, which corresponds with the measured M_s_. From the XPS spectra of the MNPs-NDA-pH8, we can only see a signal corresponding to Fe(III). There is no indication of the presence of Fe(II) on the surface of the MNPs. These results indicate that the REDOX reaction between the catechols and Fe(III) did not take place. Magnetite is composed of iron ions that are in the 2+ and 3+ oxidation states. The lack of Fe(II) dismisses the hypothesis of magnetite formation on the surface of the MNPs.

The second possible explanation for the increase in M_s_ is an increase in the spin ordering in the surface layer [1,2,10,11]. In order to ascertain whether the formation of a monolayer of catechols affects the magnetic properties of the surface layer, DFT calculations for the adsorption of CAT were made. The hydroxylated surfaces of α-Fe_2_O_3_(0001) and Fe_3_O_4_(111) were considered, as well as the two adsorption modes, as detailed in the Section 2.3. The results of these calculations are shown in Figure 6, and they reveal that the plain adsorption mode is exothermic in both cases, with a ∆*E_ads_* of −0.97 eV and −0.93 eV on the hydroxylated α-Fe_2_O_3_(0001) and Fe_3_O_4_(111) surfaces, respectively. Adsorption via condensation was found to be less favorable on both surfaces. It is almost athermic on α-Fe_2_O_3_(0001) and exothermic on the surface of Fe_3_O_4_(111), with a ∆*E_ads_* of −0.13 eV and −0.55 eV for α-Fe_2_O_3_(0001) and Fe_3_O_4_(111), respectively. Even if the latter mode is less exothermic, stronger bonds to the surface are formed during adsorption via condensation since the computed ∆*E_ads_* represents a cumulative value for both bond breaking and bond making [42]. Regardless of the adsorption mode, however, we note that the total magnetic moment of the two considered surfaces remains the same as that for the pristine surface. Indeed, the calculated total magnetization for the CAT adsorbed onto the α-Fe_2_O_3_(0001) and Fe_3_O_4_(111) is 0 μbohr/cell and 30 μbohr/cell, respectively. We can conclude that the formation of a CAT monolayer on the surfaces of MNPs does not affect the M_s_ values.

Other possible reasons for the observed increase in the M_s_ values for the functionalized particles are the size separation and the dissolution of particles [3,4,13]. The results of the size and size distribution analyses are presented in Table 2. The average size of the as-synthesized MNPs was 10.4 ± 1.2 nm, and their M_s_ was 60.0 emu/g. As can be seen from Table 2, the average sizes of the MNPs functionalized with different catechols at different pH increased, as did their M_s_. The change in size distribution is more evident if we look at the histograms, Figure 7. From these histograms, it is clear that the fraction of particles with a diameter below 10 nm is significantly lower for the functionalized MNPs. At the same time, the fraction of particles larger than 15 nm is substantially higher.

The dissolution of MNPs can affect size and size distribution. Smaller particles dissolve faster than the larger particles due to the higher surface energy. Catechols can dissolve MNPs [3] and can, therefore, affect the size and size distribution of the MNPs. The concentration of dissolved iron after functionalization with NDA at pH 8 was 0.047 μg/mL. Based on the concentration of Fe in solution, the estimated mass of dissolved MNPs was less than 0.5% of the total mass of functionalized MNPs. Although the MNPs dissolved slightly in the presence of NDA, the amount of dissolved MNPs was too small to have a substantial effect on the average size or size distribution of the MNPs.

After the functionalization, the particles were washed several times with DI water. After each washing, a small number of particles were lost. Larger particles are easier to sediment compared to the smaller ones. The smaller particles are more likely to remain in the supernatant and are washed away. The loss of smaller particles increases the average size of the MNPs, as well as the size distribution. The result is that the fraction of MNPs larger than 15 nm increases, and the total magnetization of the sample increases. This was confirmed with the control sample MNPs-pH8. M_s_ changed with the size and size distribution of the MNPs in the control tests (i.e., MNPs-pH8). We concluded that the increase in the average size explains the increase in the M_s_. The smallest effect on the size and magnetization is observed for the case when the MNPs were functionalized with NDA at pH 11. Based on the size distribution of the particles (Figure 5), we presume that the sedimentation of the MNPs, in this case, was more efficient than in the other cases.

## 4. Conclusions

In our study, superparamagnetic iron oxide nanoparticles (MNPs) were functionalized with pyrocatechol (CAT), pyrogallol (GAL), caffeic acid (CAF), and nitrodopamine (NDA) at pH 8 and 11. The functionalization of the MNPs was successful at both pH values. The thermogravimetric analyses suggested that the surface concentration of the catechols was below 3.6 molecules/nm^2^, which corresponds well with theoretical calculations for a monolayer. The exception was the functionalization of the MNPs with NDA at pH 11, where the surface concentration was 9.8 molecules/nm^2^, indicating multilayer adsorption of NDA at pH 11.

The saturation magnetization values of pure MNPs (“M_s_ pure”) were higher for the functionalized MNPs than for the starting MNPs. The increase in “M_s_ pure” could be due to the minimization of the surface-spin canting [2,8,10,14,15,16,17] or the reduction of Fe(III) to Fe(II) [12,13], followed by the formation of magnetite on the surface of MNPs. However, the results of this study show that the origin of higher magnetization is the change in size and size distribution of the functionalized MNPs. The DFT calculations showed no change in the magnetic moment after the adsorption of catechols onto the MNPs. Therefore, the adsorption of catechols did not increase the ordering of the surface spins. XPS analysis showed only the presence of Fe(III) ions in the samples, thus refuting the idea of the formation of magnetite on the surface of the MNPs. Although MNPs can be dissolved via functionalization with catechols, in our case, the dissolution was negligible. However, the size and size distribution of the MNPs changed during the process. The increased average size and a decreased fraction of MNPs smaller than 10 nm explained the increase in magnetization. Smaller particles are more difficult to sediment and are more likely to be discarded during washing. This study highlights the importance of analyzing the size of magnetic nanoparticles when evaluating their magnetic properties. When investigating the influence of ligands on the magnetic properties of MNPs, we should also consider possible changes in size and size distribution that can occur during the processing.

## Figures and Tables

**Figure 1 nanomaterials-13-01822-f001:**
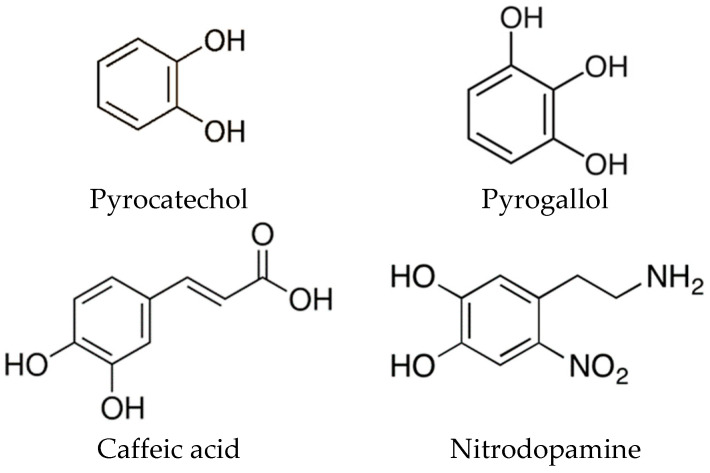
Skeletal formulas and names of the used catechols.

**Figure 2 nanomaterials-13-01822-f002:**
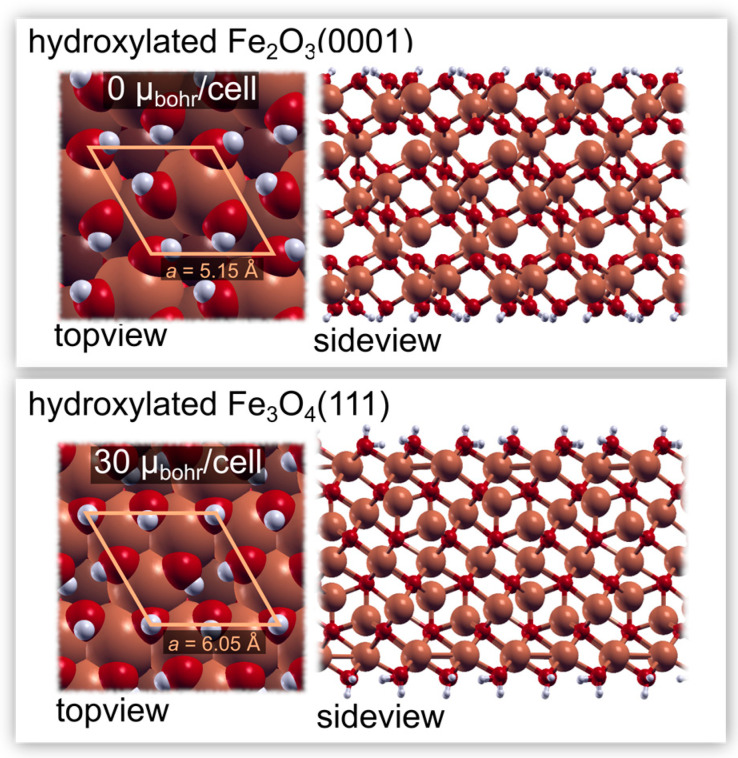
Two models of the synthesized MNP surface based on the hematite and magnetite crystalline structures. The **top panel** shows the top and side views of the fully hydroxylated Fe_2_O_3_(0001) surface, whereas the **bottom panel** shows the top and side views of the fully hydroxylated Fe_3_O_4_(111) surface. The calculated magnetization per unit cell of the periodic slabs is also stated. The fully hydroxylated Fe_2_O_3_(0001) surface is antiferromagnetic with 0 μbohr/cell. On the other hand, the fully hydroxylated Fe_3_O_4_(111) surface is ferrimagnetic with 30 μbohr/cell.

**Figure 3 nanomaterials-13-01822-f003:**
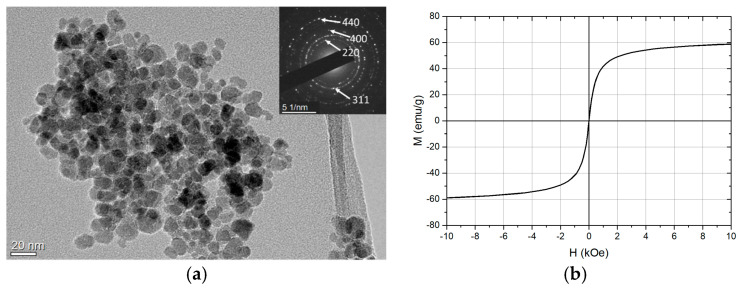
(**a**) TEM image with SAED and (**b**) magnetization hysteresis loop of bare MNPs.

**Figure 4 nanomaterials-13-01822-f004:**
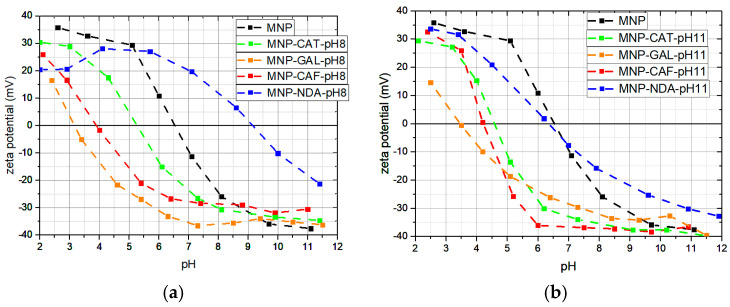
Zeta potential measurements of MNPs: bare and functionalized with catechols at (**a**) pH = 8 and (**b**) pH = 11.

**Figure 5 nanomaterials-13-01822-f005:**
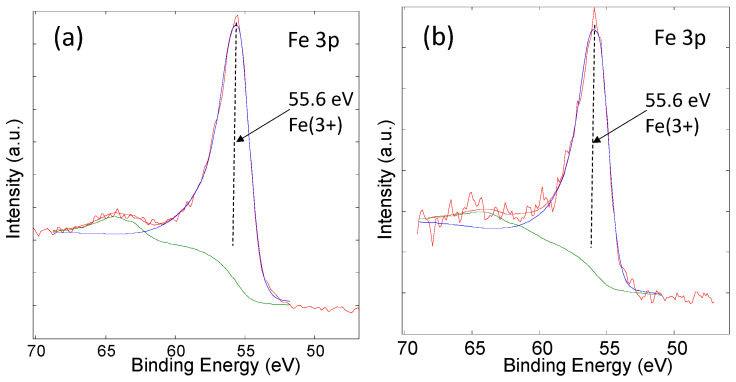
XPS spectra of (**a**) MNPs and (**b**) MNPs-NDA-pH8. The red curve is experimental, the blue curve is related to Fe(3+) oxidation state and the green curve with the satellite peak.

**Figure 6 nanomaterials-13-01822-f006:**
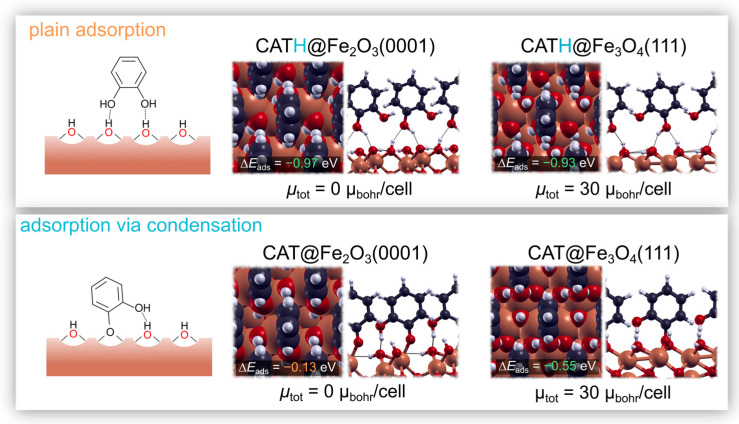
Adsorption of CAT on the fully hydroxylated surfaces of α-Fe_2_O_3_(0001) and Fe_3_O_4_(111). The **top panel** shows the structures obtained for the plain adsorption mode on the two respective surfaces, and the **bottom panel** shows the structures obtained for adsorption via condensation. The total magnetization per unit cell is also represented. Note that it remains invariant with respect to the clean surfaces (see Figure 2).

**Figure 7 nanomaterials-13-01822-f007:**
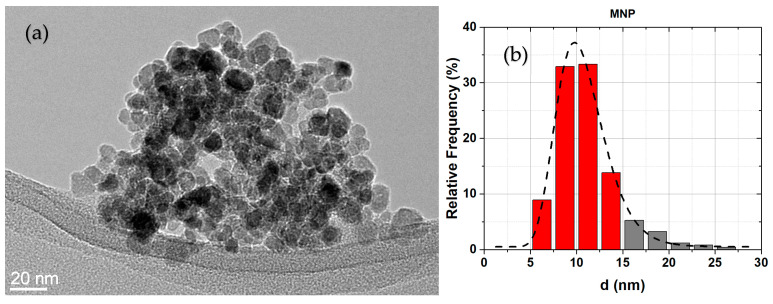
TEM images of (**a**) MNP, (**c**) MNP-NDA-pH8, (**e**) MNP-NDA-pH11, and (**g**) MNP-pH8 samples and histograms of (**b**) MNP, (**d**) MNP-NDA-pH8, (**f**) MNP-NDA-pH11, and (**h**) MNP-pH8 samples. Dashed lines are the Gaussian fit and the fractions of particles below 15 nm are highlighted with red bars to emphasize the difference between samples.

**Table 1 nanomaterials-13-01822-t001:** pK_a_ values of the used catechols.

Catechol	pK_a1_	pK_a2_	pK_a3_
CAT [22]	9.4	13.7	
CAF [23]	4.8	8.6	11.2
NDA [4]	6.5	10.0	
GAL [22]	9.05	11.2	

**Table 2 nanomaterials-13-01822-t002:** The TG loss, surface coverage, “M_s_ meas”, “M_s_ pure”, and average diameter (d) of the MNPs.

Sample	pH_ads_	TG Loss %	Surface Coverage molecules/nm^2^	“M_s_ Meas” emu/g	“M_s_ Pure” emu/g	d nm
MNP	/	4.8	/	57.1 ± 0.2	60.0	10.4 ± 1.3
MNP-CAT	8	4.8	2.9	63.2 ± 0.2	66.4	11.0 ± 1.2
MNP-CAT	11	5.9	3.6	61.5 ± 0.2	65.4	11.7 ± 1.3
MNP-CAF	8	4.0	1.5	63.4 ± 0.2	66.0	12.0 ± 1.2
MNP-CAF	11	5.4	2.0	61.6 ± 0.2	65.1	11.8 ± 1.2
MNP-GAL	8	5.4	3.2	61.3 ± 0.2	65.2	13.4 ± 1.3
MNP-GAL	11	5.9	2.9	61.9 ± 0.2	65.5	13.6 ± 1.3
MNP-NDA	8	6.4	2.3	63.6 ± 0.2	68.0	11.3 ± 1.2
MNP-NDA	11	22.7	9.8	48.4 ± 0.2	62.5	10.7 ± 1.2
MNP	8	2.5	/	62.2 ± 0.2	63.8	12.3 ± 1.3
MNP	11	4.1	/	63.6 ± 0.2	66.3	13.3 ± 0.2

## Data Availability

The data presented in this study are openly available in Mendeley Data at DOI:10.17632/3hd4rjhw36.1.

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
