# Peer review of "The Influence of Catechols on the Magnetization of Iron Oxide Nanoparticles"

_nanomaterials, 2023, doi:10.3390/nano13121822_

Round 1

Reviewer 1 Report

Reviewed manuscript could be published after major revisions. My comments are listed below:

1. Lines 213-215. "This section may be divided by subheadings. It should provide a concise and precise 213 description of the experimental results, their interpretation, as well as the experimental 214 conclusions that can be drawn" - You forgot delete it.

2. Figure 5. Why do you analyse Fe3p not Fe2p spectra?

3. Figure 7. It os not clear, how you determine size of nanoparticles.

4. Do you have any SEM or TEM images of material?

--

Author Response

1. Lines 213-215. "This section may be divided by subheadings. It should provide a concise and precise 213 description of the experimental results, their interpretation, as well as the experimental 214 conclusions that can be drawn" - You forgot delete it. -

Response to 1: The metioned text was deleted.

2. Figure 5. Why do you analyse Fe3p not Fe2p spectra?

Response to 2: Identification of the oxidation states of the Fe atoms on the surface is often performed by XPS analyses of the Fe 2p and sometimes also by the Fe 3p spectra. The Fe 2p spectra of the Fe-oxides have a complex peak shape and are difficult to identify and quantify the Fe(2+) and Fe(3+) states. In the case of the Fe-oxides, the Fe 2p spectra consist of the Fe 2p3/2 peak at about 711 eV, the Fe 2p1/2 peak at 725 eV, and also broad shake-up satellite peaks at 719 eV and 733 eV. All these peaks are present on the rather background of scattered electrons with a complex shape. Therefore, identifying the Fe(2+) and Fe(3+) states from Fe 2p spectra is complex and not simple. The alternative we used is to explore the Fe 3p spectra at the binding energy of about 55 eV. The splitting between Fe 3p3/2 and Fe 3p1/2 is very small, and the satellite peaks are much less present. Identifying the Fe(2+) and Fe(3+) states in the Fe 3p XPS spectra at binding energies of 53.8 eV and 55.6 eV is more straightforward. This was demonstrated in Reference 40 and in our previous work (Reference 41). Also, quantifying the relative concentration of the Fe(2+) and Fe(3+) states from the Fe 3p spectra may be performed.

3. Figure 7. It os not clear, how you determine size of nanoparticles.

Response to 3: The size of nanoparticles was determined from TEM images with Gatan DigitalMicrograoh as described in chapter 2. Materials and Methods/2.2 Characterization/Transmission electron microscopy

4. Do you have any SEM or TEM images of material?

Response to 4: TEM images of the materials were added.

The paper was proof read by a native English speaker and materials scientists, Dr. Paul McGuiness. Authors would appreciate, if the reviewer points out the issues related to the language. 

Reviewer 2 Report

The manuscript contains a thorough analysis of the effect of catechols on the magnetic properties of MNPs. The effect is certainly of not a great significance, but it does exist, and it is an interesting contribution to clarify the reason. Unfortunately, such s simple argument as size seems to be sufficient, but the study is quite complete and I consider the approach based on DFT calculations. I have only some rather minor observations:

1. The term “innocent” used to designate inert or indifferent compounds seems little scientific

2. The synthesis of the particles is important, and should not be reduced to just a reference. A brief account of the method used might be interesting to  the reader

3. The first three lines of the Results and Discussion section seems like instructions from the publisher. They should be deleted.

4. Page 8: why should the zeta-pH trend depend on the pH used for fucntionalization?

5. Line 279: “exhibit a signal” should read “exhibit an XPS signal” 

6. 

Author Response

1. The term “innocent” used to designate inert or indifferent compounds seems little scientific

Response point 1: The term "innocent" was replaced with "inert".

2. The synthesis of the particles is important, and should not be reduced to just a reference. A brief account of the method used might be interesting to  the reader

Response point 2: A brief description of the synthesis of particles was added.

3. The first three lines of the Results and Discussion section seems like instructions from the publisher. They should be deleted. 

Response point 3: The above mentioned lines were deleted.

4. Page 8: why should the zeta-pH trend depend on the pH used for fucntionalization?

Response point 4: The pH of functionalization affects the dissociation of the OH groups involved in the functionalization process (pKa Table 1), described in the introduction, and with this also affects the interaction mechanism between the surface iron ions and the ligand.

5. Line 279: “exhibit a signal” should read “exhibit an XPS signal” 

Response point 5: Line 279 was changed according to reviewers remark

Round 2

Reviewer 1 Report

Article could be published in present form